# The Use of Genetics and Immunology in the Diagnosis and Care of Advanced Coccidioidomycosis: Where Are We Going?

**DOI:** 10.3390/jof11090664

**Published:** 2025-09-11

**Authors:** Kavitha Thiagarajan, Shikha Mishra, Rob Purdie, Bianca Torres, Royce H. Johnson, Manish J. Butte

**Affiliations:** 1Department of Pediatrics, Division of Allergy, Immunology, and Rheumatology, UCLA, Los Angeles, CA 90095, USAmbutte@mednet.ucla.edu (M.J.B.); 2Department of Medicine, Division of Infectious Diseases, Kern Medical, Bakersfield, CA 93306, USA; shikha.mishra@kernmedical.com (S.M.); royce.johnson@kernmedical.com (R.H.J.); 3Valley Fever Institute at Kern Medical, Bakersfield, CA 93306, USA; 4Department of Medicine, David Geffen School of Medicine, UCLA, Los Angeles, CA 90095, USA; 5MyCARE Foundation, Colorado Springs, CO 80906, USA

**Keywords:** coccidioidomycosis, genetics, immunology, immune response, innate immunity, adaptive immunity, personalized treatment, immunomodulators, fungal disease, fungal diagnostics, personalized diagnosis

## Abstract

Coccidioidomycosis (CM) is an endemic disease in the western United States, northern Mexico, and Central and South America. The severity of coccidioidal infection is highly variable, with potential factors including comorbidities and the patient’s innate and adaptive immune response. Based on data from a predominantly healthy and mainly Caucasian male survey conducted nearly a century ago, approximately 60% of infections are asymptomatic, with an estimated 40% of people experiencing some respiratory symptoms; with 10% of those diagnosed with CM. Disseminated disease occurs in approximately 1% of cases and can involve the meninges and, potentially, any place in the body. It is not yet fully understood why some people experience severe disease while many people do not; it is understood that the immune response has a major role. Immunomodulators, including dupilumab and interferon-gamma (IFN-γ), have shown promise in treating patients with disseminated infection. This article summarizes the latest genetic and immunologic evidence demonstrating immune dysfunction. Immunomodulators and potential therapeutic strategies based on the above are reviewed.

## 1. Introduction

*Coccidioides immitis* and *Coccidioides posadasii* (Kingdom fungi, Phylum—Ascomycota, Class—Eurotiomecyetes, Order—Onygenales, Family—Onygenaceae, Genus—Coccidioides) are found in particular ecological sites that are widely distributed in the endemic zones [1,2,3]. There is no observed difference in the pattern of illness or outcome between the two species of *Coccidioides*. These closely related dimorphic fungi are responsible for causing the systemic infection coccidioidomycosis (CM), also known as Valley Fever. Following inhalation, arthroconidia (or spores) morph into spherules that ultimately release hundreds of endospores. These endospores then become new spherules and repeat this process [Figure 1] [4,5].

Recent modeling estimates the number of symptomatic infections as high as 273,000 annually in the USA, with the majority of cases occurring in Arizona and California [6]. Dissemination is more probable in immunocompromised patients, such as those with acquired immunodeficiency syndrome (AIDS), undergoing chemotherapy or immunosuppressive medications, and women in the third trimester of pregnancy and early postpartum [7,8,9]. Genetics may also play a role in the risk of disseminated infection [10]. Despite the above, it is difficult to predict which individuals will develop disseminated CM. CM can cause severe disease and devastating complications even in previously healthy individuals [11]. It is hypothesized that individuals with perturbations in innate and/or adaptive immune responses are more prone to develop disseminated disease [11,12,13]. Research on immunogenetics is expanding with the aim of providing personalized treatment; this is currently undergoing in a NIH-sponsored research.
Figure 1Lifecycle of *Coccidiodes* species.
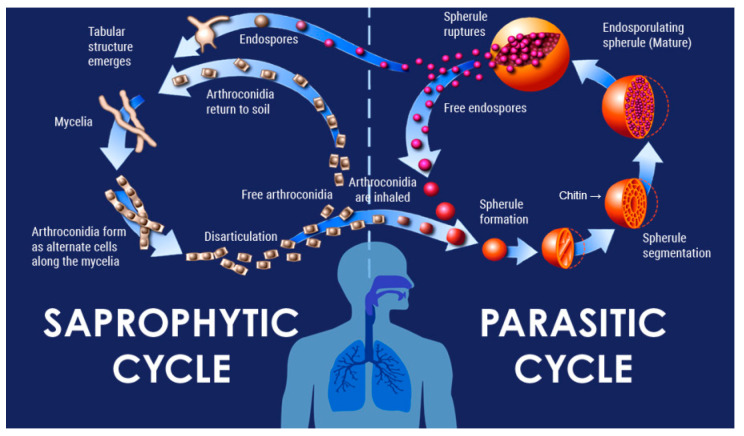


## 2. Current Diagnostics

The most common method of diagnosing coccidioidal illness is serology. Other diagnostic methods include culture, histopathology, antigen detection, polymerase chain reaction (PCR), and DNA detection tests. Histopathology demonstrating endosporulating spherules and culture of *Coccidioides* provides a definitive diagnosis. Please note that the culture of *Coccidioides* is hazardous in an unprepared laboratory. There are multiple serologic tests available that vary substantially in type, as well as in sensitivity, specificity, cost, and availability.

Lateral flow assay (LFA) tests have been studied in disseminated CM but are not readily available at most institutions [14,15]. The EIA serologic screening test detects *Coccidioides* antibodies but requires follow-up testing with immunodiffusion (ID) and or complement fixation (CF) [16]. IgG titers can be used for prognosis, to monitor response to therapy, and to diagnose relapse [17]. Many of these tests are unfortunately subjected to methodical variations across the major testing sites, which can be problematic for clinicians making care decisions. Nonspecific laboratory results, such as eosinophilia, may be a clue that the disease is not viral or bacterial, but rather coccidioidal [17,18].

## 3. Current Therapies

Antifungal therapies include azole and polyene antifungal agents. Current therapy is not fungicidal. Antifungal medications suppress the infection until the immune system is able to control it. Despite clinical resolution of signs and symptoms of disease, the fungus is still present in the body and may reactivate post-treatment. This may occur soon after therapy is discontinued or years later. Because of this, the treatment of coccidioidal meningitis and potentially other severe forms of the disease can be lifelong [9,19,20]. Control of the infection appears to depend largely on the host immune response.

Azoles are the primary therapy for less severe pulmonary, extra-meningeal disseminated, and coccidioidal meningitis [9]. Treatment failure can occur, especially in severe pulmonary and disseminated diseases. Amphotericin B liposomal therapy is used for cases that present with respiratory failure and for life- and limb-threatening extra-meningeal disease [17]. Some centers reflexively add azoles to amphotericin, while other centers switch to amphotericin exclusively. Other agents are also being studied, including olorofim, which was recently shown to maintain clinical stability in 41 patients with CM [21]. Lifelong antifungal treatment is indicated for patients with coccidioidal meningitis due to the high risk of relapse [9,20].

Glucocorticoids are also used, particularly in cases of hypoxic pulmonary disease and complications from coccidioidal meningitis [22,23,24,25]. It is important to note that administration of steroids to CM patients is only performed in conjunction with an effective antifungal agent. We endeavor to limit prolonged exposure to systemic steroids to avoid further immune suppression. Steroids are an important tool when the immune response is injuring the host, primarily in respiratory failure, which is an example of limited attempts to modulate the immune response and offer personalized care.

## 4. Immune Response

The normal immune response to fungi is complex, primarily triggered by the innate immune system, including pulmonary epithelium, neutrophils, and macrophages; the role of humoral immunity is limited [26,27]. The response is initiated by the sensing of fungal cell wall components, fungal DNA, and other fungal elements by pattern recognition receptors [26]. Recognition of beta glucan (β- glucan) by Dectin-1, a transmembrane pattern recognition receptor with an extracellular C-type lectin domain, is the first step in a signaling cascade following fungal exposure [11,28,29]. Excellent reviews have been published regarding the normal immune response to fungi [30]; here, the focus is on the mechanisms of type 2 skewing.

Cellular adaptive immunity, specifically CD4+ T helper (Th) cells, is responsible for protection against disseminated fungal disease, largely through the actions of Th1 cells [26]. Th1 cells produce interferon-gamma (IFN-γ) and augment killing by macrophages and other innate immune cells. While skewing to Th2 cells is commonly associated with morbidity in atopic conditions, it is also associated with increased morbidity and mortality in fungal infections. Parasites and fungi are pathogens known for eliciting predominantly Th2 responses, achieved through several mechanisms listed below [27].

First, it has been demonstrated that chitin, a molecule produced by both parasites and fungi (but not present in vertebrates) contributes to this Th2 response [27]. This response is driven by the recognition of chitin by chitotriosidase, a type of mammalian chitinase initially found in hosts with cryptococcal infection [27]. The recognition of chitin ultimately leads to activation of CD11b+ IRF-4 dendritic cells, which promote the recruitment of antigen-specific Th2 cells to the lung following infection [27].

Additionally, fungi and other respiratory allergens contain proteases that can degrade the tight junctions between epithelial cells in the airway, leading to compromised function of the epithelial barrier and increased permeability [31]. Since fungal components can move across the epithelial layer more easily, alarmin cytokines, including IL-25, IL-33, and thymic stromal lymphopoietin (TSLP), are released by the epithelial cells in response [31]. In the airway specifically, full-length IL-33 is cleaved by the RIPK1-caspase 8 ripoptosome in response to proteases, generating and releasing active IL-33 [31]. These alarmins activate group 2 innate lymphoid cells (ILC2s), which subsequently secrete type 2 cytokines including IL-4, IL-5, IL-9, and IL-13 to promote type 2 inflammation and activate Th2 cells [31].

Toll-like receptor 4 (TLR4) has also been implicated in type 2 skewing, although the mechanism remains unclear [32]. While most commonly known as a receptor for bacterial endotoxins, TLR4 was found to promote fungal virulence in one murine study. In this study by Dang et al., Cpl1 was identified as an effector protein secreted by the fungus *Cryptococcus neoformans*. Cpl1 drove alternative activation (M2 polarization) of macrophages by acting as a ligand for TLR4 and ultimately amplifying sensitivity of macrophages to IL-4 signaling (as opposed to type 1 immunity) [32]. While not found in the CM genome to date, Cpl1 is an example of specific proteins made by fungi that can skew to type 2 immunity.

## 5. Immune Evaluation

Comprehensive immune evaluation should be considered for all patients with disseminated CM because it is unknown what predisposes to disseminated CM; the assumption is that there must be some immune defect, whether inborn or acquired. Multiple centers and laboratories are involved in research to evaluate the immunological response to CM. The goal of this work is to identify which patients may benefit from immunological evaluation and targeted therapies. This evaluation should include complete blood count with differential, flow cytometry (to evaluate subsets of lymphocytes), immunoglobulin levels (including IgE, IgG, IgA, and IgM), cytokines (such as C-reactive protein, IL-6, CXCL9, IFN-γ, TNF-alpha, IL-4, IL-5, and IL-13), Th1/Th2 skew (if available), and functional testing of the IL-12/IFN-γ pathway to evaluate for Mendelian Susceptibility to Mycobacterial Disease (MSMD), with the understanding that the same genetic and immunologic factors that drive severe mycobacterial disease are also likely to drive severe fungal disease [11].

CXCL9 is a marker of IFN-γsignaling and is a useful test to evaluate type 1 immunity. Eosinophilia, IgE, and type 2 cytokines including IL-4, 5, and 13 are the main markers of type 2 immunity. Eosinophilia and elevated IgE have both been correlated with increased severity of CM and increased risk of dissemination [33,34,35].

The interpretation of these tests and dosing of immunomodulatory therapies is discussed below. The goal at present is to personalize treatment and augment or redirect the patient’s immune response [12,26,36]. This personalized treatment is not uncommon for patients in the oncology setting but has yet to be utilized broadly for fungal or other infectious diseases.

Overall, there is a lack of solid biomarkers of immune responses to invasive fungal infections. Criteria have not been established to evaluate whether particular biomarkers alone or in consort provide sensitive or specific revelation about the quality of an immune response. The goal in the future is to be able to better risk stratify patients.

## 6. Genetic Testing

Genetic testing is another key component of evaluation in these severe cases. It is crucial in identifying defects that can predispose patients to increased susceptibility to fungal infections. It often influences the clinical plan and can help elucidate why patients are so ill; it can also be useful in family planning.

At least a dozen monogenic immunodeficiencies leading to susceptibility to fungal infections have been identified in the past two decades. In particular, defects of the IL-12/IFN-γ pathway and Th17-mediated response are associated with increased susceptibility to disseminated fungal infections [37,38]. Disorders in this pathway are collectively referred to as MSMD. In countries outside of the United States where the BCG vaccine is routinely administered in infancy, patients with MSMD often present with early disseminated BCG infection (BCGosis) [38]. However, in the US, these patients may go undiagnosed until they present with severe fungal or mycobacterial disease later in life.

Regardless of the assumption that dissemination implies immunocompromise, disseminated CM without secondary immunocompromise has only rarely been attributed to monogenic immunodeficiency. This lack of genetic attribution is due to a number of factors including inadequate knowledge of all the genes and pathways involved in host defense against fungi; lack of testing for somatic mosaic disease; lack of testing for epigenetic defects; and many other reasons. Typically, at UCLA, patients with disseminated CM undergo either whole exome or whole genome sequencing. A recent study conducted at the National Institute of Allergy and Infectious Diseases performed whole exome sequencing on 67 patients with disseminated CM [11]. Two patients had haploinsufficient STAT3 mutations, while 51% had defects in sensing and responding to β-glucan due to variants in the Dectin-1 pathway (including variants in *CLEC7A, PLCG2,* and *DUOX1/DUOXA1*). They found that these patients had significantly lower TNF-α and H2O2 production in response to β-glucan stimulation. The production of IFN-γ, IL-12p70, and IL-17 and response to lipopolysaccharides were normal [11]. Another recent study involving 22 patients with disseminated CM found that 5 patients had genetic variants in CHIT1, encoding chitinase. Chitinase impairs the spherule formation by the fungi. It was hypothesized that impaired chitinase release by macrophages and neutrophils allowed the dissemination of CM [26].

Taken together, these results show that genes of innate and adaptive immunity are important, but more work is needed to address the gaps in knowledge about genetic conditions that confer susceptibility to disseminated CM.

## 7. Immunomodulation

Once this comprehensive immune evaluation is completed, considering immunomodulatory treatments to optimize the balance of type 1 and type 2 immunity may be warranted. These therapies are being used off-label; regardless, this approach to immunomodulation in patients with disseminated CM has proven successful [12,36,39]. When patients have received maximum antifungal treatment and are still progressing in their disease, IFN-γ should be considered. The purpose of treatment with IFN-γ is to augment type 1 immunity [26], especially neutrophils and macrophages, in their defenses. The ongoing clinical condition must be closely monitored in these patients, and the decision to continue treatment should be reconsidered at every opportunity.

Serum levels of the cytokine CXCL9 are a marker of IFN-γ signaling and are part of the initial immune evaluation described above. In patients with a low or modestly elevated CXCL9 at baseline (such as 1–2 times the upper limit of normal), subcutaneous administration of IFN-γ at a dose of 50 mg per meter squared (mcg/m^2^) three times weekly has been utilized to treat patients at UCLA. A small to moderate increase in CXCL9 indicates that the patient’s IFN-γ signaling pathway is intact, but the patient may require additional exogenous IFN-γ. Of course, considerations about their clinical condition are paramount to the decision to increase the dose. Reductions in CRP are often seen when the IFN-γ dose is sufficient. Doubling the dose (starting with 100 mcg/m^2^ three times weekly), and then tripling it (150 mcg/m^2^ three times weekly), may be warranted if the patient is not clinically improving and if the CXCL9 is not significantly raised. Typically, UCLA does not go above the 150 mcg/m^2^ three times weekly dosing. If there is no elevation in CXCL9 after starting IFN-γ, consider defects with the IFN-γ receptor or downstream signals and evaluate further with genetic testing as above.

For patients with elevated baseline CXCL9 (such as 3–4 times the upper limit of normal), it is necessary to exercise more caution if choosing to start exogenous IFN-γ. Elevated CXCL9 suggests active IFN-γ signaling. One needs to consider the risk of hemophagocytic lymphohistiocytosis (HLH) if additional IFN-γ is given. Cases of HLH in the setting of CM have previously been published, highlighting the importance of caution when using IFN-γ with elevated CXCL9 [40]. In these cases, consider checking other inflammatory markers such as ferritin, closely monitor for evidence of any developing cytopenias, and continue trending CXCL9.

The use of systemic IFN-γ treatment has been described as an adjunctive therapy in several reports of treatment-refractory disseminated disease [36,39,41]. In summary, starting IFN-γ should be strongly considered in patients with disseminated CM refractory to antifungals who have evidence of low type 1 immunity.

Another potential therapeutic agent is dupilumab, a monoclonal antibody that binds to the alpha subunit of the IL-4 receptor, antagonizing IL-4 and IL-13 signaling [42]. The goal of treatment with dupilumab is to suppress type 2 immunity in patients with Th2 skewing [26]. Blocking Th2 immunity increases Th1 immunity, though Th1 is in direct balance not only with Th2, but also with Th17, induced T regulatory cells, and other helper T cell patterns. Nonetheless, recent data have shown that when IL-4 and Th2 signaling is blocked with dupilumab, skewing to Th1 occurs [3]. Additionally, the higher the patient’s baseline IL-4, the more their type 1 signaling with IFN-γ was recovered upon treatment with dupilumab. These data were recently presented at the Clinical Immunology Society 2025 Annual meeting, and a manuscript detailing these findings is currently being prepared for publication [36].

Dupilumab has been used for disseminated cases of CM in children and has shown a promising response [36]. It was first successfully used in the treatment of a 4-year-old patient with disseminated CM complicated by extensive multifocal osteomyelitis. His disease was refractory to antifungals, and the bony involvement slowed but did not resolve with exogenous IFN-γ [12]. After starting dupilumab, the patient’s lytic lesions, inflammatory markers, and IgE all improved. Additionally, dupilumab was successful in improving clinical outcomes in a case series of 10 patients with disseminated disease refractory to maximum antifungal therapy and type 2 skewing at UCLA [36]. Dupilumab should be considered in patients with disseminated CM refractory to antifungals and with evidence of type 2 skewing, such as eosinophilia and elevated IgE. At UCLA, cytokines produced by activated helper T cells are examined to adjudicate Th2 skewing. The dosing of dupilumab is similar to that for the treatment of asthma, depending on the patient’s age and weight.

The duration of immunomodulatory therapies has been determined on an individual basis; trending the Th1/Th2 skew and type 2 cytokines (IL-4, 5, 13) after starting immunomodulatory treatment has been useful. Since the Th1/Th2 assay is not commercially available and each patient is unique, the authors are willing to collaborate in the care of these complex cases.

## 8. Discussion

This paper briefly provides an overview of the current issues and opportunities with individualized evaluation and care of advanced CM. The path forward could address fundamental defects in the diagnosis and care received by patients. A superior understanding of an individual’s genetic and immunologic background that contributed to their symptomatic/severe CM would advantage them with potential new therapies. Personalized care using immunologic and genetic testing can lead to additional therapeutic options and potentially better outcomes for patients.

While the current costs of immunologic testing and immunomodulating therapies may be prohibitive for most institutions, costs for these new technologies have been shrinking as the platforms become more common and the therapies become available as biosimilars and generics. The cost of caring for patients with disseminated CM is high—in the US, billions of dollars are spent annually on patients with CM [43,44,45]. While the exact cost/benefit ratio is unclear, there is definitely a need for faster, lower cost assays for immune evaluations in these severely ill patients. Ideally, this need will be the impetus for the development of lower cost assays and treatments.

In the initial sections, we address the current state of the art in coccidioidomycosis, followed by a discussion of future directions to optimize care of personalized diagnosis.

## 9. Conclusions

Ultimately, there is an unmet need for the personalized evaluation and treatment of multiple diseases, multiple infections, and certainly cases of disseminated CM. Current diagnostics and therapeutics are solely aimed at the pathogen and are found wanting in many cases.

Comprehensive immune evaluation, genetic testing, and adjuvant immunomodulatory treatments should be considered for these complex cases. Further research is needed to better elucidate genetic and immunologic pathophysiologic factors predisposing to disseminated CM.

## Data Availability

Not applicable.

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
