# Peer review of "The Use of Genetics and Immunology in the Diagnosis and Care of Advanced Coccidioidomycosis: Where Are We Going?"

_jof, 2025, doi:10.3390/jof11090664_

Round 1

Reviewer 1 Report

Comments and Suggestions for Authors

The Use of Genetics and Immunology in the Diagnosis and Care of Advanced Coccidioidomycosis: Where Are We Going?

As the title indicates, this is an ambitious topic.  Unfortunately, this review has major inadequacies and in places is misleading. 

  • Section 1 (Introduction) has many errors. See specific items below.
  • Section 2 (diagnosis) is too brief to be useful and also contains misstatements.
  • Section 3 (current therapies) is also too brief and could be deleted since it is unrelated to the genetics and immunology purpose of this review.
  • Section 4 (immune response) Much of this draws on what is known about other fungi and parasites. This may or may not be involved in host response to Coccidioides.
  • Section 5 (immune evaluation) This is a laundry list of immunologic analyses which the authors suggest should be considered for all patients with disseminated coccidioidomycosis. However, the authors state “Overall, there is a lack of solid biomarkers of immune responses to invasive fungal infections. Criteria have not been established to evaluate whether particular biomarkers alone or in consort provide sensitive or specific revelation about the quality of an immune response.” If this is the case, then the immunologic evaluations suggested here really should be performed in a research setting rather than in routine care of patients with disseminated infection.
  • Section 6 (genetic testing) Similar concern that this is an important area for research but not likely to be useful for routine clinical care.
  • Section 7 (Immunomodulation) Line 190 citations 2, 3, 38 cite three case reports from two peer-reviewed papers and a Conference abstract. Reference 40 cited later adds an additional case report. The reference I have suggested below adds additional experience with interferon gamma, both positive and negative. While these authors are to be commended for trying to develop immunomodulation therapies, it seems premature to describe this approach as anything but experimental. I would suggest that the section be rewritten with this concern in mind and have it be part of the “Where Are We Going” that is in the title.

Specific items that need attention:

Lines 23-24: Add reference 3a: https://doi.org/10.1093/cid/ciaf286

Line 36, provide a reference here more recent than 1938.  Many to choose from.

Line 38, replace reference 7 with doi:10.1001/jamanetworkopen.2025.13572 and edit the text to conform with this report’s findings.

Line 40-41, Diabetes does not predispose to disseminated infection. Reference 10 demonstrates an association of diabetes with cavitation, not dissemination.  Remove diabetes from this sentence and remove reference 10.

Line 41-42, Genetic ancestry is a meaningless term. This sentence adds nothing to this manuscript.

Lines 44-46, references should be [2, 13, 15].  Reference 14 has to do with CSF serology.

Line 53, culture is “also definitive” rather than “confirmatory.”

Lines 55-56, there is only one LFA test commercially available, that marketed by IMMY.  In addition to reference 14, it has been studied in early coccidioidal infections (https://www.ncbi.nlm.nih.gov/pubmed/32818956)

Lines 65-66, this statement is wrong as ref 19 demonstrates.  The immune system does not control cocci meningitis when antifungal therapy is stopped.

Lines 89 and 174, protein names should be in all CAPS so DECTIN-1 not Dectin-1

Lines 162-163, the statement that IL12/IFNg pathway patients in the US go undetected in childhood is misleading. A patient with IFNg receptor and disseminated cocci had an atypical mycobacterial infection as a baby (https://www.ncbi.nlm.nih.gov/pubmed/19681704). A patient with a IL12r mutation had cervical adenopathy due to Salmonella .(http://www.ncbi.nlm.nih.gov/pubmed/21258095). 

Author Response

Comments 1: Lines 23-24: Add reference 3a: https://doi.org/10.1093/cid/ciaf286

Response 1: Thank you for bringing this to our attention. The other reviewers suggested that there should not be references in the abstract, so all references have been removed.

Comments 2: Line 36, provide a reference here more recent than 1938.  Many to choose from.

Response 2: Agree. We have added the reference https://doi.org/10.1080/21505594.2019.1589363. Please see line 39 and reference 4.

Comments 3: Line 38, replace reference 7 with doi:10.1001/jamanetworkopen.2025.13572 and edit the text to conform with this report’s findings.

Response 3: Thank you for pointing this out. We have updated the symptomatic infections in line 42 and the reference, please see reference 6.

Comments 4:Line 40-41, Diabetes does not predispose to disseminated infection. Reference 10 demonstrates an association of diabetes with cavitation, not dissemination.  Remove diabetes from this sentence and remove reference 10.

Response 4: Thank you for pointing this out. Diabetes has been removed (see line 45-46) and reference 10 has been removed.

Comments 5: Line 41-42, Genetic ancestry is a meaningless term. This sentence adds nothing to this manuscript.

Response 5: The authors strongly feel this sentence connects to the subject of the paper and believe it to be an important line. The term has been changed from “Genetic ancestry” to “Genetics”.

Comments 6: Lines 44-46, references should be [2, 13, 15].  Reference 14 has to do with CSF serology.

Response 6: Thank you for citing this. The references have been updated

Comments 7: Line 53, culture is “also definitive” rather than “confirmatory.”

Response 7: Thank you for this suggestion. We have updated the test to include histopathology and culture as definitive. This has been updated in the text, please see line 58.

Comments 8: Lines 55-56, there is only one LFA test commercially available, that marketed by IMMY.  In addition to reference 14, it has been studied in early coccidioidal infections (https://www.ncbi.nlm.nih.gov/pubmed/32818956)

Response 8: Thank you for this, in addition to reference 14, the above article has been added as reference 15, please see line 63.

Comments 9: Lines 65-66, this statement is wrong as ref 19 demonstrates.  The immune system does not control cocci meningitis when antifungal therapy is stopped.

Response 9: Thank you for this note. The original text may have been unclear, and the authors did not mean to say that coccidioidal meningitis is controlled when therapy is stopped. We have modified and added sentences in lines 73-76 for clarification.

Comments 10: Lines 89 and 174, protein names should be in all CAPS so DECTIN-1 not Dectin-1

Response 10: Thank you for bringing this to our attention. We looked into this and Dectin is not in all caps in most articles. In references 28 and 29, his is not written in all caps. Because of this, Lines 109 and 195 have remained the same as Dectin-1.

Comments 11: Lines 162-163, the statement that IL12/IFNg pathway patients in the US go undetected in childhood is misleading. A patient with IFNg receptor and disseminated cocci had an atypical mycobacterial infection as a baby (https://www.ncbi.nlm.nih.gov/pubmed/19681704). A patient with a IL12r mutation had cervical adenopathy due to Salmonella .(http://www.ncbi.nlm.nih.gov/pubmed/21258095). 

Response 11: Thank you for bringing this to our attention. In line 183, the authors did not make this an absolute sentence as we said “may go undiagnosed…”

Reviewer 2 Report

Comments and Suggestions for Authors

Dear authors;

The manuscript has academic and scientific relevance; however, I suggest the following adjustments.

  1. It is not common to add references to the abstract; please remove them.
    2. Add a diagram (figure) to the introduction that demonstrates the mycosis transmission cycle to enrich the manuscript.
    3. Locate, in the Eukarya domain classification, where these fungi are inserted (family, order, etc.).
    4. Line 40: Indicate the meaning of AIDS and add that this information is valid when the patient is untreated HIV-positive.
    5. Line 57: Italic format.
    6. Line 100: I suggest adding a figure of the fungal cell wall, indicating the presence of chitin. There is no figure or table in the manuscript. Why?
    7. Lines 159-163: The authors point out a situation that occurs in the USA regarding BCG vaccination. This also occurs in Brazil. Please provide more information about other countries that do and do not adopt this vaccination protocol and the consequences of this event on the epidemiology of this fungal disease in those countries.
    8. What is the economic feasibility (costs) of a diagnosis based on genetic testing? Do the authors have this data? Could this be applied in low-income countries or only in developed countries like the USA?
    9. The discussion raises the cost of diagnosis based on immunomodulatory therapies; however, the discussion on this topic is superficial. Please add other studies that corroborate or better discuss the topic presented in the manuscript.
    10. Improve the manuscript's conclusion.

Kind regards.

Reviewer#

---

Author Response

Comments 1: It is not common to add references to the abstract; please remove them.

Response 1: Thank you for this suggestion. The references have been removed from the abstract.  

Comments 2: 2. Add a diagram (figure) to the introduction that demonstrates the mycosis transmission cycle to enrich the manuscript.

Response 2: Thank you for this suggestion. Are there specific instructions on how to add the diagram below? I’ve included it

Comments 3: 3. Locate, in the Eukarya domain classification, where these fungi are inserted (family, order, etc.).

Response 3: Thank you for this suggestion, this has been added, please see lines 31-32.

Comments 4: 4. Line 40: Indicate the meaning of AIDS and add that this information is valid when the patient is untreated HIV-positive.

Response 4: Thank you for this suggestion, this has been updated, please see like 45.

Comments 5: 5. Line 57: Italic format.

Response 5: Thank you for this, this has been updated.

Comments 6: 6. Line 100: I suggest adding a figure of the fungal cell wall, indicating the presence of chitin. There is no figure or table in the manuscript. Why?

Response 6: Please see the diagram.

Comments 7: 7. Lines 159-163: The authors point out a situation that occurs in the USA regarding BCG vaccination. This also occurs in Brazil. Please provide more information about other countries that do and do not adopt this vaccination protocol and the consequences of this event on the epidemiology of this fungal disease in those countries.

Response 7:

Comments 8: 8. What is the economic feasibility (costs) of a diagnosis based on genetic testing? Do the authors have this data? Could this be applied in low-income countries or only in developed countries like the USA?

Response 8: Currently not economically feasible but may well be in the future in developed countries.

Comments 9: 9. The discussion raises the cost of diagnosis based on immunomodulatory therapies; however, the discussion on this topic is superficial. Please add other studies that corroborate or better discuss the topic presented in the manuscript.

Response 9: References for these are 43-45

Comments 10: 10. Improve the manuscript's conclusion

Response 10: Thank you for this suggestion, we have updated the conclusion.

Reviewer 3 Report

Comments and Suggestions for Authors

It presents a truly novel topic: the need to direct research efforts toward personalized diagnosis and treatment for disseminated coccidioidomycosis, as well as many other diseases. However, I have some observations regarding the presentation of the work:

The abstract should not include references. It should include a summary of the entire content of the work, including conclusions.

Lines 47, 263-264, 246, 247, 250, please include the corresponding references.

Keywords: It is recommended not to include words that are in the title.

In the diagnosis section (2 Diagnosis), the authors present a summary of the most commonly used techniques, but they do not offer an opinion on which direction diagnosis should take, focusing on "personalized diagnosis." I find this relevant, since the conclusion of the work highlights the need for personalized evaluation and treatment in many diseases, including disseminated coccidioidomycosis.

Author Response

Comments 1: The abstract should not include references. It should include a summary of the entire content of the work, including conclusions.

Response 1: Thank you for this suggestion. The other reviewers suggested that there should not be references in the abstract, so all references have been removed.

Comments 2: Lines 47, 263-264, 246, 247, 250, please include the corresponding references

Response 2:  

Comments 3: Keywords: It is recommended not to include words that are in the title.

Response 3: Thank you for this suggestion. We reviewed other manuscripts and they include words that are in title so this has not been changed.

Comments 4:In the diagnosis section (2 Diagnosis), the authors present a summary of the most commonly used techniques, but they do not offer an opinion on which direction diagnosis should take, focusing on "personalized diagnosis." I find this relevant, since the conclusion of the work highlights the need for personalized evaluation and treatment in many diseases, including disseminated coccidioidomycosis.

Response 4: Thank you for this suggestion. While we do agree with this, this section is to focus on where we are with diagnostics and not an opinion on which direction diagnosis should take, as further sections discuss this. We have updated the “Diagnosis” section to “Current Diagnostics”.

Reviewer 4 Report

Comments and Suggestions for Authors

This paper is devoted to coccidioidomycosis, an endemic disease in the western United States. The main content of the review is devoted to the consideration of traditional and promising diagnostic methods. The work has a number of significant shortcomings that negatively affect the quality of the manuscript. First of all, the structure of the manuscript seems unjustified: in particular, it is unclear why the section "Diagnostics" is followed by a discussion of drug therapy and immune response, after which the authors return to discussing the diagnosis of the disease. The sequence of sections should be changed.The authors do not pay enough attention to the content, the material they discuss appears incomplete and requires significant revision. Some of the authors' statements are not fully disclosed.

1. There is a lack of information on the biology of the pathogen. The authors provide virtually no data on the characteristics of the pathogens being discussed, which limits the reader's understanding and deprives them of context. There is no information on the lethality of the disease, risk groups, or characteristics of the course. There is also no justification for the relevance of developing new diagnostic methods.

2. In Section 2, "Diagnostics," more attention should be paid to disclosing the problem of diagnostics: what is the effectiveness of each method, which of them are preferable and in what cases. The authors (lines 49-52) also mention genetic studies, but do not disclose the content of this diagnostic approach. The authors mention two pathogens, it should be indicated whether there is a difference in the frequency and effectiveness of detecting each of them.

3. Section 3 "Current Therapies" needs to be improved: similar to what was previously announced, the authors do not pay enough attention to the disclosure of important details. It is necessary to supplement the section with information on the preferred drug therapy for coccidioidomycosis, on the limitations and potential solutions to the risks that arise. There is also a complete lack of information on the risks of developing drug resistance, which largely determines the success of therapy for fungal infections.

4. In section 4, despite the authors' position on the role of immunodeficiency states on the course of the disease, there is no mention of how exactly the infection can develop depending on the immune status or concomitant chronic diseases. Also, ideas about Th1/Th2 imbalance are duplicated in sections 4, 5 and 7 without adding new information.

5. It would be interesting to know what are the differences in the course of coccidioidomycosis between patients with focal infection and developed dissemination. I think it is important to demonstrate a similar difference in relation to both pathogens. Is there a difference in lethality, the rate of infection spread, or the mechanisms of the immune response?

6. Sections 5-7 are devoted to the use of genetic information and immunology for successful diagnosis of diseases, but the authors do not sufficiently disclose what advantages in-depth diagnostics of patients opens up for clinicians. In the Discussion, the authors say that the benefit of such diagnostics is not obvious, but they do not talk about the disadvantages, except for the high price, and the advantages. The authors should carefully review these sections, supplement them with new information and especially with arguments for their position.

7. Dupilumab is approved for the treatment of atopic dermatitis, asthma, and other Th2-dependent diseases, but not for coccidioidomycosis. On the other hand, IFN-γ (strongly recommended in the article) is produced by a limited number of companies. Notifying this would help readers assess the objectivity of the recommendations. The authors should provide this information. Also, the use of an off-label drug may raise a conflict of interest: the authors should clearly clarify this point.

8. In section 7, the authors recommend IFN-γ and dupilumab, but do not discuss potential side effects (e.g., risk of HLH with IFN-γ) or limitations of dupilumab use (e.g., in patients with concomitant helminthiasis). Overall, the authors do not pay enough attention to the risks of immunomodulation. 9. The article is positioned as an "Opinion", but contains data from an unpublished study (link to the abstract of CIS 2025 (3rd source in the list), line 233-234). This information needs to be revised.
10. In Section 9, the authors state an "unmet need" for diagnostic tools, but do not offer specific directions for future research.

Reading this work raises more questions than it answers. I would also like to note that the text of the manuscript is inconsistent and fragmentary. For example, in lines 129-130, the authors state the goal of the study - however, this occurs practically at the end of the work and excludes the importance of subsequent sections. The authors also work quite carelessly with academic language and often do not explain the meaning of their statements, for example, line 150.

To summarize, the article does not correspond to the level of the journal, requires a serious revision of the content and data provided. The authors must do a lot of work so that this manuscript can be published. In its current form, the article cannot be recommended for publication.

Author Response

Comments 1: 1. There is a lack of information on the biology of the pathogen. The authors provide virtually no data on the characteristics of the pathogens being discussed, which limits the reader's understanding and deprives them of context. There is no information on the lethality of the disease, risk groups, or characteristics of the course. There is also no justification for the relevance of developing new diagnostic methods.

Response 1: Thank you for this suggestion. Another reviewer suggested we add the transmission lifecyle to the manuscript and figure 1 had been added after the introduction. The risk groups are outlined in lines 44-47.  And the lethality characteristic has been mentioned in the abstract with justification made in lines 47-48.

Comments 2: 2. In Section 2, "Diagnostics," more attention should be paid to disclosing the problem of diagnostics: what is the effectiveness of each method, which of them are preferable and in what cases. The authors (lines 49-52) also mention genetic studies, but do not disclose the content of this diagnostic approach. The authors mention two pathogens; it should be indicated whether there is a difference in the frequency and effectiveness of detecting each of them.

Response:  Thank you for this response. We have added a sentence about the two pathogens, please see line 34-35.  This diagnosis section was meant to give a brief overview of current diagnostics. We have updated the section title to “Current Diagnostics” as after this section, we are headed to immune diagnosis of the host and immune therapy of the host.

Comments 3: 3. Section 3 "Current Therapies" needs to be improved: similar to what was previously announced, the authors do not pay enough attention to the disclosure of important details. It is necessary to supplement the section with information on the preferred drug therapy for coccidioidomycosis, on the limitations and potential solutions to the risks that arise. There is also a complete lack of information on the risks of developing drug resistance, which largely determines the success of therapy for fungal infections.

Response 3:  Thank you for this suggestion. Line 89-90 mentions primary therapy for which kinds of cases. Line 91-93 mentions when to use liposomal amphotericin B, line 98-99 mentions the role of glucocorticoids in hypoxic and coccidioidal meningitis, we can elaborate significantly on the treatment but the authors think the purpose of the article is to focus on advanced diagnostics like genetics and immunotherapy. Treatment- if patients have side effects or fail 1 azole therapy its recommended to try a second azole vs liposomal, depending on the institution and clinical judgement.

Comments 4: 4. In section 4, despite the authors' position on the role of immunodeficiency states on the course of the disease, there is no mention of how exactly the infection can develop depending on the immune status or concomitant chronic diseases. Also, ideas about Th1/Th2 imbalance are duplicated in sections 4, 5 and 7 without adding new information.

Response 4:  Thank you for this suggestion. We talk about Th1/Th2 imbalance in multiple sections to help

Comments 5: 5. It would be interesting to know what are the differences in the course of coccidioidomycosis between patients with focal infection and developed dissemination. I think it is important to demonstrate a similar difference in relation to both pathogens. Is there a difference in lethality, the rate of infection spread, or the mechanisms of the immune response?

Response 5: Thank you for this suggestion. Lines 17-20 discuss the differences in the course of the disease. There is no observed difference in the pattern of illness or outcome between the two species. We have added this to likes 34-35.

Comments 6: 6. Sections 5-7 are devoted to the use of genetic information and immunology for successful diagnosis of diseases, but the authors do not sufficiently disclose what advantages in-depth diagnostics of patients opens up for clinicians. In the Discussion, the authors say that the benefit of such diagnostics is not obvious, but they do not talk about the disadvantages, except for the high price, and the advantages. The authors should carefully review these sections, supplement them with new information and especially with arguments for their position.

Response 6: Thank you for this comment. We have addressed the discussion and conclusion sections and have updated them accordingly. We want to point out that the potential is to develop immunoadjuvant therapies that is individualized.

Comments 7: 7. Dupilumab is approved for the treatment of atopic dermatitis, asthma, and other Th2-dependent diseases, but not for coccidioidomycosis. On the other hand, IFN-γ (strongly recommended in the article) is produced by a limited number of companies. Notifying this would help readers assess the objectivity of the recommendations. The authors should provide this information. Also, the use of an off-label drug may raise a conflict of interest: the authors should clearly clarify this point.

Response 7: Thank you for this comment. Line 237-244 explains the limitations of using IFNy and the need to use other modes of therapy.  There is no unlabeled drug approved for coccidioidomycosis.

Comments 8: 8. In section 7, the authors recommend IFN-γ and dupilumab, but do not discuss potential side effects (e.g., risk of HLH with IFN-γ) or limitations of dupilumab use (e.g., in patients with concomitant helminthiasis). Overall, the authors do not pay enough attention to the risks of immunomodulation.

Response 8: Thank you for this comment. The authors feel this is not a paper about implementing specific therapeutic but evaluating the host/parasite paradigm.

Comments 9: 9. The article is positioned as an "Opinion", but contains data from an unpublished study (link to the abstract of CIS 2025 (3rd source in the list), line 233-234). This information needs to be revised.

Response 9:

Comments 10: 10. In Section 9, the authors state an "unmet need" for diagnostic tools, but do not offer specific directions for future research.

Response 10: Thank you for this comment. The authors have updated the conclusion and the discussion section to address the topic.

Round 2

Reviewer 1 Report

Comments and Suggestions for Authors

n/a

Reviewer 2 Report

Comments and Suggestions for Authors

Dear authors;

Thanks for changes in the manuscript.

kind regards.

—-

Reviewer 4 Report

Comments and Suggestions for Authors

The authors have substantially corrected the manuscript in accordance with the comments. I have no more significant comments and propose accepting the revised manuscript for publication.